# Views from Multinational Pharmaceutical Companies on Allocation of Clinical Trials in Saudi Arabia—Qualitative Study

**DOI:** 10.3390/pharmacy12060167

**Published:** 2024-11-12

**Authors:** Nouf M. Aloudah, Ahmed M. Shaman

**Affiliations:** Department of Clinical Pharmacy, College of Pharmacy, King Saud University, Riyadh 11523, Saudi Arabia; shaman@ksu.edu.sa

**Keywords:** clinical trials, qualitative study, interviews, Saudi Arabia, pharmaceutical studies

## Abstract

Clinical trials conducted by pharmaceutical companies are essential for bridging local research efforts with broader populations, facilitating the transfer of valuable insights and solutions. This study aimed to explore the barriers and facilitators affecting clinical trials in Saudi Arabia from the perspective of key personnel within the pharmaceutical industry and Contract Research Organizations (CROs). We conducted in-depth semi-structured interviews with nine participants, which provided a holistic understanding of the intricate dynamics shaping the landscape of clinical trials in the country. The analysis revealed three prominent themes: operational challenges, complexities in navigating approval hurdles, and the unique value proposition for conducting clinical trials in Saudi Arabia. The participants expressed pride in the local infrastructure but acknowledged existing flaws, particularly in regulatory processes that contribute to delays in trial initiation. They emphasized the importance of conducting clinical trials in areas such as diabetes, crowd management during pilgrimages, and rare diseases, which are prevalent in the region. Despite the limited number of clinical trials registered (354 from 2009 to 2020, with only 1% being phase 1 studies), Saudi Arabia’s total pharmaceutical market exceeds SAR 13 billion, positioning it as the largest market in the region. Stakeholders recognized the country’s potential as a research hub, particularly within the Gulf Cooperative Council (GCC) region. However, to attract more trials and enhance the medical research landscape, it is crucial to address the identified barriers, streamline processes, and improve stakeholder alignment. The findings highlight the need for targeted interventions to overcome these challenges and leverage Saudi Arabia’s investments in healthcare infrastructure since its transformation program launched in 2010. By enhancing the regulatory environment and fostering collaboration among stakeholders, Saudi Arabia can solidify its role as a key player in international clinical research.

## 1. Introduction

Research plays a pivotal role in expanding our understanding and knowledge of various issues, particularly in the realm of healthcare. In this regard, the clinical trials conducted by pharmaceutical companies serve as a cornerstone, providing invaluable expertise and resources that contribute to the advancement of medical knowledge and the fight against diseases. Moreover, these trials play a crucial role in bridging local research efforts with the broader population, thereby facilitating the transfer of valuable insights and solutions to a wider audience. Furthermore, understanding how drugs work and their clinical effects, including benefits and side-effects, within the Saudi population offers immense value, particularly when considering the significant role of pharmacogenomic aspects in tailoring personalized treatments. Additionally, the involvement of pharmaceutical companies in clinical trials brings with it not only expertise but also essential clinical equipment and technical tools. These resources not only facilitate the development of health-related expertise but also contribute to the improvement in healthcare practices, ultimately leading to better health outcomes for individuals and communities [1].

Multinational pharmaceutical companies prioritize recruitment-related factors when allocating clinical trials [2]. These companies often fund the majority of clinical studies [3]. Decision-making processes within these companies involve coping strategies to manage non-linear and evolving situations in clinical trial work [4]. Additionally, there is a preference among multinational pharmaceutical companies to conduct trials in their home regions [5].

Pharmaceutical companies often outsource research and clinical trials to Contract Research Organizations (CROs) [6], and they focus on sustainability investments to enhance visibility and increase company value [7]. The allocation of clinical trials by these companies is influenced by various factors, including recruitment considerations, decision-making strategies, and preferences for trial locations. Initiatives promoting transparency and data sharing aim to address ethical concerns and rebuild trust between pharmaceutical companies and the public.

In conducting clinical trials, pharmaceutical companies often encounter various regulatory challenges, particularly in resource-limited settings. These challenges include complex and politically motivated regulatory frameworks, which can lead to significant delays in trial approvals, as seen in countries like Zambia, India, and South Africa [8]. There is also a lack of continuous monitoring from regulatory bodies, affecting data integrity and participant safety. Moreover, the emergence of data transfer agreements and conflicting policies between funders and in-country regulators can complicate data management and publication rights. Additionally, the registration and shipping of new drugs face hurdles due to limited local expertise and logistical issues. Addressing these challenges requires harmonizing regulatory processes, enhancing collaboration between international and local regulators, and improving the capacity and expertise of regulatory bodies. Understanding these barriers is crucial for enhancing Saudi Arabia’s position as a preferred site for conducting clinical trials by pharmaceutical companies.

In a systematic review, Alemayehu et al. (2018) [9] examined the barriers encountered in conducting clinical trials within developing countries, where the burden of both communicable and non-communicable diseases is disproportionately high. This review identifies five primary themes that impede research efforts: lack of financial and human resources, ethical and regulatory system obstacles, inadequate research environments, operational barriers, and competing demands on healthcare professionals. Despite the urgent need for context-specific medical research, this review highlights a significant under-representation of developing countries in the global clinical trial landscape, compounded by challenges unique to these settings, such as local structural and procedural issues. The authors advocate for the establishment of local investigator-initiated trials, positing that such studies are not only more relevant to local populations but also align with national health strategies, ultimately fostering a more equitable distribution of health research benefits. This study contributes valuable insights to the literature on healthcare research methodologies in low-resource contexts, emphasizing the necessity for tailored strategies to enhance clinical research capacity in developing nations.

Clinical research in Saudi Arabia is limited; for example, in a descriptive study conducted for all phases of clinical trials registered between 2009 and 2020, there were only 354 clinical trials found, of which 44 and 1% were phase 4 and 1 studies, respectively. This number is small for a population of more than 30 million in Saudi Arabia. Factors such as secure funding, available expertise, and a lack of laws and regulations were suggested to contribute to such results [3].

Saudi Arabia’s total pharmaceutical market is more than SAR 13 billion (USD 3.51 billion), making it the largest market in the region. Allocating clinical trials in Saudi Arabia could bring significant benefits, but it requires careful planning and resources. Interventions addressing reward innovation, speeding up patients’ access, fully protecting intellectual property, and improving regulatory approval might be needed. Exploring the barriers and facilitators for conducting clinical trials sponsored by pharmaceutical companies in Saudi Arabia and mapping the situation might help inform better intervention(s) design and decision making. After a search of the literature, no previous study explored the factors of these limitations and how to improve them. Thus, our aim is to explore barriers to the conduct of pharmaceutical companies’ clinical trials in Saudi Arabia from the perspective of these companies and to identify recommendations to improve Saudi Arabia’s position as a preferred site for conducting clinical trials.

### Objective

Exploring barriers to the conduct of pharmaceutical companies’ clinical trials in Saudi Arabia from the perspective of these companies.Identify recommendations to improve Saudi Arabia’s position as a preferred site for conducting clinical trials.

## 2. Methods

### 2.1. Study Design

This study implemented an in-depth semi-structured qualitatively designed study consisting of in-depth interviews conducted with key personnel involved in pharmaceutical companies’ clinical trials in Saudi Arabia.

### 2.2. Participates and Setting

A search was conducted to identify pertinent documents released by governmental and private stakeholders in Saudi Arabia, encompassing regulatory, ethical, research, and development agencies. These documents were analyzed thematically to understand the primary barriers that potentially contribute to the limited engagement of pharmaceutical companies in clinical trials in Saudi Arabia. The insights gleaned from this analysis were instrumental in enriching the developed topic guide (see Appendix A), which served as a foundational framework for delving deeper into these barriers during the in-depth interview study. By integrating these findings into the topic guide, this study was able to explore and address the identified barriers in a comprehensive and focused manner during the interactions with the study participants.

### 2.3. Document Analysis

Documents related to conducting pharmaceutical companies’ clinical trials carried out in Saudi Arabia were collected using a diverse array of sources, including the internet and various open-access repositories, involving PubMed and Google Scholar to access scholarly articles, as well as exploring pharmaceutical companies’ websites for relevant clinical trial information, to search for specific terms related to factors influencing clinical trial allocation in Saudi Arabia. The search strategy involved entering the following keywords—clinical trial allocation, patient recruitment, pharmaceutical companies, trial site selection, and Saudi Arabia—and retrieving their related terms.

Data were extracted from the selected documents using a predesigned Excel spreadsheet (Appendix A) where each row is a document and each category is the information that we want to extract. The categories were author, date, topic/title, institution/source, source, barriers, and recommendations. Throughout this process, the researcher critically engaged with the material, aligning the extraction process with the research questions to construct a holistic understanding of the contributing factors. The findings were used to supplement the development of the in-depth interview topic guide.

### 2.4. In-Depth Interview

The top 10 pharmaceutical companies conducting clinical trials in Saudi Arabia were contacted. We employed a snowball sampling method to identify knowledgeable contacts within pharmaceutical companies. This involved initial contacts recommending other individuals who possessed relevant expertise on the subject, ensuring that we engaged with the right people who could provide valuable insights. The interviews were conducted by Nouf Aloudah, an expert in the qualitative research field since 2014. Her extensive experience and expertise in qualitative methodologies have been instrumental in ensuring the depth and quality of the data collected. In terms of reflexivity, the author’s (NA) background as an associate professor and pharmacist, combined with her years of qualitative research experience, provided her with a nuanced understanding of the subject matter. This expertise allowed her to navigate interviews effectively, fostering an environment where participants felt comfortable sharing detailed and accurate information. The robustness of this study, including the reliability and validity of the interview materials, is further elaborated in the data analysis section below, where we outline the rigorous methods employed to ensure the integrity of our findings. Each interview lasted between 40 and 60 min. The interviews were undertaken virtually using a Zoom videocall.

### 2.5. Topic Guide

A preliminary topic guide (Appendix A) was created by the author (NA), drawing upon insights from the relevant literature. To ensure cultural relevance and contextual appropriateness in the Saudi Arabian setting, the topic guide underwent a review by two experienced pharmacists. During the interviews, the topic guide served as general guidance while allowing for flexible probing to adeptly explore the participants’ perspectives. Notably, the interviews were conducted in the English language, ensuring effective communication and comprehension among all participants.

To evaluate the rigor and trustworthiness of this study, we applied the Guba and Lincoln criteria [10], specifically focusing on confirmability, dependability, credibility, and transferability. For confirmability, which ensures the impartiality of the findings, we maintained an audit trail of the data analysis process, conducted member checking, and kept a reflexive research journal. Dependability, which addresses the consistency of a study, was supported by documenting procedures and processes through an audit trail and maintaining a reflexive research journal. Credibility, which relates to the accuracy and truthfulness of the findings, was enhanced by prolonged engagement with the subject, ensuring the referential adequacy of materials, conducting peer debriefing, and performing member checks. Finally, transferability, which considers the applicability of the findings to other contexts, was reinforced through the use of thick descriptions and purposive sampling. These strategies collectively enhance the potential applicability of this study’s outcomes in diverse contexts and with other participants.

### 2.6. Data Analysis

#### 2.6.1. Part One: Document Analysis

This study adopted a systematic approach for document analysis, following the READ approach [11], which encompassed the following sequential steps:Ready Your Materials: This initial step involved preparing and organizing the collected materials to ensure a structured and comprehensive foundation for subsequent analysis.Extract Data: Following the preparatory phase, the relevant data were meticulously extracted from the materials, focusing on identifying key information and pertinent details essential to the research objectives.Analyze Data: The extracted data were subjected to thorough analysis, employing robust analytical methods to discern patterns, themes, and insights crucial to addressing the research inquiries effectively.Distill Your Findings: In this conclusive phase, the findings derived from the analytical process were distilled, allowing for the synthesis of comprehensive and actionable insights that contribute to the broader understanding of the subject matter.

By adhering to the systematic READ approach, this study ensured a structured analysis of the collected documents, thereby facilitating a comprehensive and insightful exploration of the research domain.

#### 2.6.2. Part Two: In-Depth Interview

The thematic analysis of the interviews was conducted using MAXQDA Analytics Pro 2020 (MAXQDA Analytics Pro, version 2020, produced by VERBI Software GmbH, Berlin, Germany), a robust platform for qualitative data analysis. Each transcript underwent independent analysis by two researchers (NA, AS), with any disparities resolved through thorough discussions to ensure consensus and accuracy in the interpretation of the data. Following each semi-structured interview, dedicated sessions were held to develop initial codes and identify significant emerging information, serving as a crucial step in the iterative refinement of the analysis process and the comprehensive exploration of key themes. In a commitment to methodological rigor and trustworthiness, each interview was concluded with a summary that was validated by the participants, providing an opportunity to address any potential ambiguities and validate the accuracy of the captured information. Moreover, post-interview, researchers convened to reflect on the insights collected, fostering continuous refinement and alignment in the interpretation of the data. Memos and journaling maintained throughout the interviews were instrumental in capturing and preserving critical insights, subsequently utilized to inform the ongoing data collection and analysis within MAXQDA through the integration of memos. Notably, data security protocols were strictly adhered to, ensuring the confidentiality and integrity of the collected interview data. The data collection process continued until the point of saturation. Saturation was reached when subsequent interviews consistently reiterated previously identified themes, indicating that the data had become sufficiently comprehensive to support this study’s findings. This decision was documented and cross-verified through peer debriefing sessions to ensure its validity.

## 3. Results

This study interviewed nine key personnel of the pharmaceutical companies and Contract Research Organizations (CROs) in Saudi Arabia, and the analysis revealed three prominent themes encompassing the operational challenges encountered, the complexities of navigating approval hurdles in clinical research, and the unique value proposition for clinical trials in Saudi Arabia. The following detailed presentation captures the viewpoints of the participants on each theme, supported by relevant quotations. Figure 1 presents a summary of the views of the multinational pharmaceutical companies on the allocation of clinical trials in Saudi Arabia.

### 3.1. Theme 1: Operational Challenges

The theme of operational challenges in clinical trials in Saudi Arabia encompasses several key issues. Participants highlighted procedural difficulties, such as limited access to trial sites for monitors and challenges in comparing source notes due to security restrictions. Engaging investigators and ensuring their availability were emphasized as crucial for trial success, with governance and awareness playing significant roles. Payment challenges were also noted, as a lack of standardized incentives often demotivates staff. The absence of clinical research coordinators (CRCs) was identified as a gap, affecting the management of trials and the ability to address new obstacles. Additionally, the need for a robust data culture was stressed, with participants calling for centralized data hubs to facilitate evidence-based medicine. Overall, these operational challenges underscore the need for improved procedures, engagement strategies, and infrastructure to enhance clinical trial efficiency and effectiveness in the region.

#### 3.1.1. Dynamics of Procedures

The interviewee discussed various operational challenges faced in the clinical trial process. These challenges include difficulties in the accessibility to sites for monitors and the lack of access even to blinded data. Monitors find it difficult to complete tasks such as comparing source notes to the electronic data capture system due to security or institution staff withdrawal. To address these challenges, management should deepen their expertise to effectively overcome obstacles. These procedures can cause delays and problems, but they can be overcome:

“Yes, indeed, when entering, for example, in major institutions such as the […], a military body, for example, it’s not suitable for anyone … there must be a gate pass, for example. These are the things that barriers or delay, causing some delays and issues. However, these can be easily overcome. It’s not the big issue. These are the things that the monitors faces, which requires the PI and the coordinator to meet them. They might not be available, they might be traveling, or they might be busy with other things.” (RCTPh 5, Pos. 79).

“And also, really they challenge, one of the main challenges that I bear a large part of the responsibility for, is that they don’t go and question Investigators at the research center over site quality concerns in the clinical trials. They don’t freely quantify their performance….” (RCTPh 4, p. 3).

“Operational challenges that we face today; for example, we have sites where we have monitors. Their main job is to handle say four studies… which we call Source Notes and compare the source notes of the patients to the EDC (Electronic Data Capture). EDC is usually a third-party system where the site is feeding the patient data required by the clinical trial. It is a blinded data and doesn’t identify a patient, one of the main challenges we are currently facing is the accessibility to the sites… we had an issue with expats being able to access the sites, mainly due to security reasons…” (RCTPh 4, p. 8).

#### 3.1.2. Importance of Investigator Engagements

Furthermore, there were discussions on the availability of engaged staff with the necessary skills and experience. The institution’s governance and awareness also play a role in recruitment success. The participants emphasized the importance of quality and corrective actions in ensuring the successful recruitment and engagement of PIs:

“No, the organizations, governance, and ASOB’s exist, but as I told you, the awareness and willingness from the PIs is what needs more attention.” (RCTPh 5, Pos. 50).

#### 3.1.3. Challenges in Payments

The engagement of staff might be affected by the issue of receiving incentives for participating in clinical trials, which is a common problem. Many participants do not receive compensation, because they are considered to be doing their job as part of their hospital work. This lack of incentive can be a burden for those who do not have a strong research motivation. There is a lack of a systemized approach to incentives, as there are no standardized incentives across institutions:

“But do these amounts ultimately reach the researcher conducting the clinical trial? Most of the time, no, because they consider it as part of their work in the hospital, which makes it lacking an incentive. What motivates me to work like this? If I personally don’t have a strong research incentive, it becomes a burden on me, or I don’t have an incentive.” (RCTPh 7, Pos. 68).

“… This budget is split between the PI, sub-investigator, study coordinator, and others such as pharmacist, radiologist depending on the study, and technician, etc. However, we have sites where the head over the site or the research center, one of the main challenges we face is that the study team and investigators don’t receive their money, which demotivates them. As a Sponsor, the payment goes to the institution based on the contract. The split between the institution and the study team, and when we come to conduct another study, the study team tells you that we haven’t received our money!” (RCTPh 4, p. 16).

#### 3.1.4. Lack of Clinical Research Coordinators

Furthermore, the coordination activities of nurses, coordinators, data entry, and data exports are important in clinical trials. Administrative people like legal and documentation also play a crucial role in relieving doctors of some burdens. Clinical research coordinators (CRCs) are important in discovering new obstacles, identifying opportunities, and working with researchers:

“… is the lack and availability of clinical research monitors who monitor the site performance. These monitors are often at the level of pharmaceutical and CRO societies. We have a huge gap at the local pharmaceutical societal level” (RCTPh 4, p. 7).

#### 3.1.5. Lack of Data Culture

The participants further discussed the importance of data availability in the healthcare industry, particularly for clinical trials and evidence-based medicine. They emphasized the need for a data culture in hospitals and a centralized data hub for easy access to all relevant data:

“… but what I mean to say is that the data exists, it might be at the institution level, but do you have a one-stop shop where I can access and find all the data I need to run? Actually, it’s not available yet. There are registries for some diseases, but still, the registry is…” (RCTPh 7, Pos. 109).

### 3.2. Theme 2: Navigating Approval Hurdles in Clinical Research

#### 3.2.1. Fragmented Regulatory Landscape

The participants discussed the importance of having an authority responsible for clinical trials from start to finish. The interviewees mentioned the need for a well-organized system and the importance of having a clear order:

“It is necessary that there should be someone, I mean a person with authority, responsible for the clinical trials from A to Z, and has the authority to solve the issue …. So, when we talked to the different authorities there, it is well, but it needs a push, even the well needs to be in seized…” (RCTPh 7, Pos. 81).

#### 3.2.2. Concerns and Qualifications of Institutional Review Board (IRB) Members

The participants further discussed the challenges faced by the pharmaceutical industry in terms of capabilities and resources, particularly in relation to IRB expertise. As discussed, the issue of IRB expertise is twofold, one at the level of sites and the other at the level of researchers. To address this challenge, there is a need to create a new generation of healthcare practitioners who have the capability to participate in decision making and become IRB experts:

“Sure, as far as I know… the capabilities when it comes to clinical practices are very strong, but I have other capabilities related to the processes. For example, the ideal profile should be present in the internal review report in the IRB so that they can make a quick and sound decision, ask the right questions? This is not very clear now; it depends on connections and relationships. So, this is one of the things we said the capabilities is to create a second generation or batches of doctors or healthcare practitioners who have the ability to participate in such decisions and become IRB experts. This is a very important matter.” (RCTPh 7, Pos. 120–121).

#### 3.2.3. Challenges in Obtaining Approval from Several Authorities

Furthermore, receiving approvals involves several steps that need to be shortened; for example, one participant discussed the process of obtaining approval for research involving the collection and transportation of samples. The process involves obtaining approval from the Institutional Review Board (IRB) and filling out forms from the National Committee for Bioethics (NCPE) and the Saudi FDA. Approval from the Zakat and Tax Authority (ZADAKA) is also necessary for transporting the samples. The interviewees mentioned that obtaining approval from ZADAKA can take varying amounts of time:

“… the process involves obtaining IRB approval, then there is the form for the NCPE. If you are aware of this NCPE, it is filled out and signed by the IRB and comes with IRB approval for the NCBE form. Then, we take this form and send it to the NCPE, send it to the Saudi FDA, and send it to the Zakat and Tax Authority, which is responsible for customs at the airport. It’s just for information for the NCPE and the Saudi FDA, but I need approval from the Zakat Authority to export the samples. I need approval to export the samples. I have two trials, one of which I obtained in 48 hours, and the other in two and a half months personally.” (RCTPh 6, p. 7–8).

### 3.3. Theme 3: Unique Value Proposition for Clinical Trials in Saudi Arabia

#### 3.3.1. Long-Term Benefits and Impact of Clinical Trials

The participants discussed local benefits of the pharmaceutical industry, including conducting clinical trials to advance scientific knowledge through understanding the local population:

“… but it’s rich in clinical trials. In my personal opinion, it’s one of the very good things, scientific knowledge and scientific interaction with doctors in the country. This means that when a new drug is launched today and doctors try it in the clinical trials, they can provide actual real-life insights and what happens when they use it with their patients during that clinical trial. This is something I’ve seen in [drug name] … [drug name] is very significant globally, but [type of virus] is the most common in the Middle East, however, it’s very small compared to globally. They always talk about [other virus type]. However, this is a different thing, every recommendation in the dialogue, two different viruses, so it’s the sites locally and what we have seen, for example, Japan always has its own trials because the population there is different from the rest of the world. The same goes for China, as there are things that make a difference when you have a doctor locally try the drug, understanding what’s good, understanding what’s bad, understanding how to use, understanding the recommendations, all of these things make a difference. For example, it will make a difference when you have discussions, when you train the other doctors … the protocol, our drugs, local data, all of these things, meaning the investment we put into the clinical trial, the benefits, the read, we will also see in a year, two years, three years, five years, whatever, when the drug is actually available, and we can use this discussion” (RCTPh 6, p. 13–14).

#### 3.3.2. Pride in Infrastructure

The participants mentioned the availability of impressive infrastructure in Saudi Arabia, which visitors are amazed by during tours. However, they also acknowledged that there are flaws:

“… I told you about all the drawbacks, but not all the advantages. We have advantages, for example, infrastructure. Whenever you go and see […] building, you will be amazed by the equipment and the buildings and so on, right When visitors from abroad come, we give them tours, and they are amazed by these things…” (RCTPh 5, Pos. 188–192).

#### 3.3.3. Disease Diversity as a Research Hub

The participants discussed the concept of a unique value proposition for their pharmaceutical company in Saudi Arabia. They consider which medical fields would be most beneficial to conduct clinical trials in, such as diabetes and rare diseases, due to the high prevalence of these conditions in the population. They also mention the importance of conducting clinical trials related to mass gatherings, such as the Hajj and Umrah, in order to prepare for potential pandemics:

“Let me tell you something else, that I can find, as they say, a unique value proposition for Saudi Arabia. As a pharmaceutical company, if I conduct clinical trials in ten fields, which field should I consider right away for clinical trials in Saudi Arabia? For example, oncology? Maybe, but not strong, because the truth is that oncology is more common in later age groups, and we have a population were, for example, diabetes might be more suitable due to the large number. What about rare diseases, like the less common diseases? opportunity? Clinical trials related to mass gatherings like Hajj and Umrah, as happened in Covid, mean we need to conduct clinical trials on this model…” (RCTPh 7, Pos. 99).

#### 3.3.4. Delay and Its Consequences

The participants discussed the negative impact of entering late and the implications for various stakeholders:

“… Sometimes recruitment is competitive, for example, if a study requires, let’s say, 100 patients globally, and if you are late, another country committed to, for example, ten patients, then entered 20, they would take them away from you. As a result, you could end up with zero, and they would win. It’s like a game between us and them.” (RCTPh 5, Pos. 108).

He further mentioned that our country has problems affecting our reputation. We need better alignment with stakeholders to attract clinical trials from the outside:

“… but when we encountered these problems, our image was somewhat affected externally by these things that are supposed to be well-organized … we want to portray a bright image of our infrastructure, the quality, and so on, but things like this come up, which can affect our reputation externally. No, we need to invest more in alignment, better alignment with all the stakeholders so that our country is an attractive destination for clinical trials from outside.” (RCTPh 5, Pos. 104).

## 4. Discussion

This study interviewed key personnel in the pharmaceutical industry in Saudi Arabia, and this approach facilitated a holistic understanding of the intricate dynamics shaping the landscape of clinical trials within the Saudi Arabian pharmaceutical industry, uncovering insights into the value proposition for conducting clinical trials in the country. Participants expressed pride in the local infrastructure but acknowledged existing flaws. They also discussed disease diversity as a research hub, emphasizing the importance of conducting clinical trials in medical fields such as diabetes, crowd management (pilgrimage), and rare diseases due to their high prevalence in Saudi Arabia. Furthermore, they highlighted the impact of delayed entry on the country’s reputation and the need for better stakeholder alignment to attract trials from the outside. Saudi Arabia has become a prominent hub for research, particularly in the field of medicine. The country has significantly invested in its healthcare sector, launching a transformation program in 2010 to enhance its infrastructure and capabilities [12]. This focus on healthcare development has positioned Saudi Arabia as a key player in medical research, with a growing emphasis on disease diversity and clinical trials. Stakeholders in the pharmaceutical industry have acknowledged the importance of conducting clinical trials in Saudi Arabia. The country’s potential as a research hub is further supported by its leading role in clinical trial productivity within the Gulf Cooperative Council (GCC) region [13]. To sustain this progress, it is essential to address existing flaws, streamline processes, and improve stakeholder alignment to attract more trials and advance medical research in the country.

Our results highlight the multifaceted dynamics within the Saudi Arabian pharmaceutical industry regarding the conduct of clinical trials, shedding light on the local infrastructure and the diverse disease landscape as critical strengths. Participants recognized the country’s advancements in healthcare, aligning with the findings of Alemayehu et al.’s (2018) systematic review [9], which identified the importance of establishing a conducive research environment in developing nations for enhancing clinical trial activities. While our study emphasizes the pride in local infrastructure coupled with the acknowledgment of existing flaws, Alemayehu et al. pointed out that operational challenges, such as lengthy regulatory processes and inadequate research conditions, can significantly impact trial conduct in developing countries. Additionally, the focus on diseases pertinent to Saudi Arabia, including diabetes and rare diseases, resonates with the call for local investigator-initiated trials suggested by Alemayehu et al., which are more aligned with national health priorities and can drive sustainable outcomes. Furthermore, the necessity for improved stakeholder alignment in our findings reflects the broader call for collaboration noted by Alemayehu et al., who emphasized the need for synergy among researchers, regulators, and healthcare providers to optimize clinical trial participation and effectiveness.

Furthermore, the pharmaceutical industry faces challenges related to the regulatory landscape and ethical concerns surrounding Institutional Review Board (IRB) expertise. Participants in the industry have emphasized the need for a well-organized system and clear authority responsible for overseeing clinical trials. Obtaining approval from multiple authorities poses challenges, and addressing issues related to IRB expertise at both site and researcher levels is crucial [14,15].

The involvement of for-profit Contract Research Organizations in clinical research has increased, leading to concerns about financial conflicts of interest and the potential impact on research integrity [14]. Additionally, the reliance on commercial IRBs rather than academic institutions for regulatory oversight has raised questions about the adequacy of ethics training and the monitoring of ethical practices in clinical trials [16].

Multicenter studies face challenges in obtaining IRB approval, necessitating the development of strategies to streamline the review process [15]. The varying responses of IRBs to standard protocols highlight the need for greater consistency and efficiency in the review process [17]. Moreover, the ethical considerations surrounding the collection, use, and storage of human biological samples for research underscore the importance of clear regulatory guidance and ongoing training for IRB members [18].

Additionally, operational challenges discussed in this paper included difficulties in institutions’ gate passes and data access, the importance of investigator engagement, challenges in payments, and a lack of clinical research coordinators. The participants emphasized the need for a data culture in hospitals and a centralized data hub for easy access to relevant data. Investigator engagement plays a pivotal role in the success of clinical trials, influencing participant recruitment, retention, and adherence to protocols. Challenges in payments can deter clinicians from participating in trials, underscoring the importance of fair compensation to enhance recruitment and retention rates [19].

Establishing a data culture within hospitals is essential for promoting data-driven decision making and ensuring that research activities are supported by robust data infrastructure. A centralized data hub can streamline data access, enhance collaboration among stakeholders, and facilitate efficient data sharing for research purposes [19].

To substantiate the significance of the findings of our study, it is imperative to propose concrete operational measures aimed at improving the efficiency and ethical conduct of clinical trials conducted by pharmaceutical companies in Saudi Arabia. This research aims to form part of a broader implementation research initiative, designed to delve into the complexities of current clinical trial operations. By identifying barriers and facilitators within existing systems, this study aims to inform better planning and execution of future interventions. The focus on implementation research not only addresses immediate operational challenges but also contributes to the sustainability and ethical integrity of clinical trials in the region [8]. Future work should involve further investigation and development of specific operational measures that can be effectively applied in real-world settings, thereby bridging the gap between theoretical insights and practical application.

This study’s findings are based on interviews with key personnel involved in pharmaceutical companies’ clinical trials in Saudi Arabia, which may limit the generalizability of the results due to the sample size and diversity. The qualitative nature of the research means the findings are context-specific and may not apply to other regions with different regulatory environments. Document analysis was constrained by the availability of relevant materials, potentially affecting the comprehensiveness of the empirical knowledge. Potential bias could arise from participants’ personal experiences and the researcher’s interpretation of themes, despite efforts to ensure reflexivity. Additionally, the focus on pharmaceutical company-sponsored trials may overlook challenges faced by academic or governmental trials. Finally, the evolving regulatory environment means the findings may need to be revisited as new guidelines are implemented.

Our study highlights several operational challenges faced by pharmaceutical companies and Contract Research Organizations (CROs) in Saudi Arabia, which align with broader regulatory issues observed in other resource-limited settings [8]. Participants noted the complexities of navigating approval hurdles, echoing the difficulties described in the literature where changes in regulatory frameworks and politically motivated regulations lead to delays and increased resource allocation difficulties [8]. Additionally, the need for continuous monitoring of clinical trials to ensure data integrity and participant safety is a concern, as inconsistent monitoring standards can hinder trial progress and outcomes [9]. This study also emphasizes the unique value proposition Saudi Arabia holds for clinical trials, suggesting potential for improvement by addressing these regulatory challenges. Harmonizing review processes and enhancing collaboration between local and international regulators could improve Saudi Arabia’s position as a preferred site for clinical trials, as suggested by the broader context of international regulatory challenges [9]. Integrating insights from global experiences with local regulatory developments could support the maturation of Saudi Arabia’s drug regulatory authority, facilitating efficient and timely clinical trial approvals and reinforcing the country’s value proposition in the global clinical research landscape.

In conclusion, this study highlights the complex landscape of clinical trials in Saudi Arabia, focusing on the value proposition, approval challenges, and operational issues faced by pharmaceutical companies. The findings provide insights that could help improve the efficiency and ethical conduct of clinical trials in the region. Specifically, there is a need for better alignment among stakeholders, more streamlined regulatory processes, and enhanced operational support. These insights contribute to a better understanding of the clinical research environment in Saudi Arabia and may inform future efforts to advance this field.

## Figures and Tables

**Figure 1 pharmacy-12-00167-f001:**
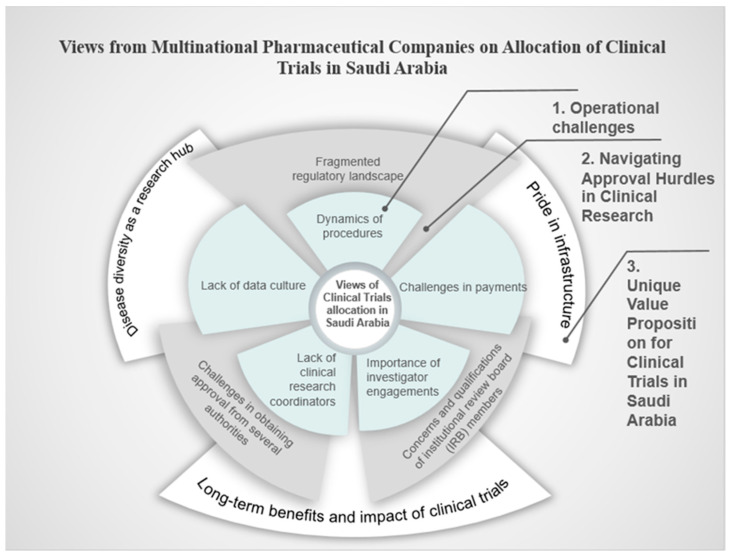
The views of the multinational pharmaceutical companies on the allocation of clinical trials in Saudi Arabia.

## Data Availability

The data supporting the reported results of this study are available upon request from the corresponding author.

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
