# Peer review of "Views from Multinational Pharmaceutical Companies on Allocation of Clinical Trials in Saudi Arabia—Qualitative Study"

_pharmacy, 2024, doi:10.3390/pharmacy12060167_

Round 1
Reviewer 1 Report
Comments and Suggestions for Authors
Dear Authors,
After careful evaluation of the manuscript "Multinational Pharmaceutical Companies Views of Clinical 2 Trials allocation in Saudi Arabia- Qualitative Study ", you can find below my comments.
- I recommend a more comprehensive literature review in the Introduction paragraph on this topic to provide scientific support for the results obtained;
- The Results and Discussion section presents the results of the study. However, it is important that this section also includes an interpretation of the results and a comparison with several similar studies in the literature;
- The results of the article may be of interest to a range of academic disciplines. However, in order to be considered for publication, it is recommended that the research be complemented by further research (proposing concrete operational measures to improve the efficiency and ethical conduct of clinical trials of pharmaceutical companies in Saudi Arabia) to substantiate the significance of the findings.
- I recommend the inclusion of a larger number of more recent studies on the subject and their insertion in bibliographic references;
- I recommend a more concise and thorough description of the data collected, their interpretation and the experimental conclusions that can be drawn;
- I recommend that conclusions be formulated which emphasise the importance of the results obtained, as this paragraph is missing.
Based on the above mentioned, I recommend this paper for publication after performing the suggested correction.
Best regards,
Reviewer 2 Report
Comments and Suggestions for Authors
See attached copy. A review of English should be performed and help with grammatical issues. I would use barriers instead of hindrances or hinders. Watch using words such as Moreover, Furthermore.
Many aspects are overstated or stated with too much confidence. Keep to the results as found.
The methods section needs more information on the interviews. Please describe these methods in detail. Who did the interviews, how were the questions derived, and did just one person do them (were they trained?)?
In the results, quotes are used from subjects 4,5,6,7 --what about the others? I did not see a true thematic analysis with results from all interviews. I think your conclusion gives the themes clearly, but how was this derived?
You report no limitations to the study, but there are limitations to discuss. Those must be reported.
3.1 Theme 1 needs expanded with more explanation.

The article needs some significant revisions related to the English language.
Reviewer 3 Report
Comments and Suggestions for Authors
Elaborate on the key findings and make them more tangible in the abstract. You may want to delete some repeated phrases.
Address grammatical issues and improve sentence structure for more formal and academic tone.
It’s not hinders, it’s barriers.
Methodology section is robust.
Discussion: While the challenges surrounding IRBs are discussed, there is no mention of potential solutions. The discussion would benefit from suggestions on how to overcome these hurdles
Conclusion: The concluding sentences are vague. The phrases are a repetition of earlier points without offering new insights or strong takeaways. The conclusion should summarize the key points more explicitly and perhaps suggest actionable recommendations or areas for future research.
Highlight specific interventions or solutions that could emerge from the study as a part of the conclusion.
Comments on the Quality of English Language
- There are several grammatical errors, such as "this number is small", instead they would have used "this number is notably low").
- Sentences like "Allocating clinical trials in Saudi Arabia will brings great benefit but requires tackling limited aspects" contain errors ("brings" should be "bring" and "limited aspects" is vague). Several such minor grammatical errors were defined.
- The sentence structures in some parts feel informal